# Evaluation of a nursing and midwifery exchange between rural and metropolitan hospitals: A mixed methods study

Amy-Louise Byrne[1]*, Clare Harvey[1], Diane Chamberlain[2], Adele Baldwin[1], Brody Heritage[3], Elspeth Wood[1]

1 School of Nursing, Midwifery and Social Sciences, Central Queensland University, Townsville, Queensland, Australia, 2 College of Nursing and Health Sciences, Flinders University, Adelaide, South Australia, Australia, 3 Murdoch University, Murdoch, Western Australia, Australia

* a.byrne@cqu.edu.au

**Data Availability Statement:** All relevant data are within the paper and its Supporting Information files.

## Abstract

### Introduction

This paper reports on the findings of the Nursing and Midwifery Exchange Program, initiated to promote rural and remote nursing and midwifery, and to facilitate clinical skills development and clinical collaboration between health services in Queensland, Australia. The project was undertaken over an 18-month period in one state of Australia, offering structured, temporary exchange of personnel between metropolitan and rural health services.

### Background

Globally, there is an increasing awareness of nursing shortages, and with it, the need to ensure that nurses and midwives are prepared for specialist roles and practice. This is particularly important in rural and remote areas, where there are pre-existing barriers to access to services, and difficulties in attracting suitably qualified, permanent staff.

### Methods

A mixed methods approach to the evaluation was undertaken with two cohorts. One cohort was the nurses and midwives who participated in the exchange (n = 24) and the other cohort were managers of the participating health services (n = 10). The nurses and midwives who participated in the exchange were asked to complete a questionnaire that included questions related to embeddedness and job satisfaction. The managers participated in a Delphi series of interviews.

### Results

Those who participated in exchange reporting a higher score on the reported degree of understanding of rural client, which was accompanied with a moderate-to-large effect size estimate ($d = 0.61$). Nurses/midwives in the exchange group reported higher scores on their perceptions of aspects of their home community that would be lost if they had to leave, which was accompanied with a large effect size ($d = 0.83$). Overall, NMEP was reported by

**Funding:** Funding for this project was provided by Queensland Health.

**Competing interests:** The authors have declared that no competing interests exist.

the participants to be a positive way to improve professional development opportunities for nurses and midwives. The findings also show the program supported practical collaboration and raised the profile of nursing and midwifery in rural areas.

## Conclusion

Exchange programs support clinical and professional development, raising the awareness of different contexts of practice and related skills requirements, and thereby supporting a greater understanding of different nursing roles. In the light of increasingly complex care required by patients with chronic conditions being managed in community-based services, programs such as NMEP provide the opportunity to build collaborative networks between referring and referral centres as well as contribute to the ongoing skills development.

## Introduction

The Nursing and Midwifery Exchange Program (NMEP) was designed as an innovative workforce solution in response to continued challenges associated with recruitment of rural and remote nursing and midwifery staff, as well as maintaining professional development and promoting interest in rural/remote nursing and midwifery careers. Funded through the Queensland Office of the Chief Nursing and Midwifery Office (OCNMO), the program was developed and maintained through the South West Hospital and Health Service (SWHHS) Queensland, Australia. However, all sixteen Hospital and Health Services (HHS) across the state of Queensland were eligible to engage staff. The program itself aimed to 'match' a rural/remote nurse or midwife with a metropolitan/regional nurse or midwife and a professional job swap was facilitated. Candidates could choose to exchange over three or six months, with most candidates choosing the three months option. Candidates engaging in the exchange were employed and paid at their host facility and were eligible for up to AUS$5000 additional funds for travel and accommodation. Permanent Queensland Health Registered and enrolled nurses and midwives in at least their second year of practice were eligible to apply. NMEP commenced in August of 2017 with the project completing in June 2019. A total of 46 candidates completed the exchange.

## Background

Like many Organisation for Economic Cooperation (OECD) countries, Australia is facing a shortage of nurses, brought about by the ageing workforce set to retire within 10 to 15 years [1]. This is of concern, given that nursing forms the largest health workforce in Australia [2]. As chronicity and an ageing population increases, so too are the push/pull effects of acute hospitals transferring patients back to communities earlier in their recovery than in the past. Nursing preparedness and the nursing skill mix to meet the needs of the higher acuity patients being managed by community and regional health services [3] is subsequently of higher importance. Moreover, the challenge is in attracting nurses to rural areas and keeping them there, with rural nursing and midwifery not often seen as an inspiring career pathway [4, 5]. Australia has vast geographical areas serviced by regional and remote health services. The more remote the service, the more likely it is to be significantly larger in size, with notably less staff [6]. This adds to the challenge for nurses working in rural locations, who are frequently the sole practitioners, with primary responsibility to provide effective care, supported by

remotely located medical practitioners and specialist services [7]. The challenges related to recruiting and retaining a suitably qualified rural and remote nursing health workforce are also reflective of a broader international problem.

Innovative strategies take many approaches, including the use of technology to address the challenges of distance, and as such, technology such as telehealth services is well utilised in some rural and remote areas. However, some of the regional and remote locations in Australia do not have access to this technology [8], with some not having access to reliable internet services upon which telehealth relies. Nursing in these areas thus forms the mainstay of rural and remote health care, requiring multi-purpose skills to support care from the cradle to the grave, making their repertoire of skills broad but in a sense, also specialised [9].

Australia's Primary Health Strategy identifies the value of nurses and their contribution to a changing model of care that sees the point of service embedded in community health services [10]. This is particularly important to nursing in rural and remote areas, considering the complexity of care required for people living with chronic conditions. As the primary providers of care, nurses are often the professionals who help patients navigate the multiple services and specialities they require, across multiple regional and metropolitan centres [11]. Integrated care thus, has been described as essential, although the value of such care is reliant on an effective co-ordinator to manage it [12, 13], which rural and remote nurses are proving to be effective at [14–16]. Greater transferability of the nursing workforce and the preparedness for managing care in a dynamic health care environment, provided within the framework of programs such as NMEP, become essential.

The principle idea of NMEP was to provide opportunity for exposure to different nursing contexts with a view to promoting regional areas of the state, raising awareness of nursing and midwifery in regional areas of Queensland. The objective was to build a more sustainable nursing and midwifery workforce through the collective strength of the state's resources to support and develop the skills of the nursing workforce across regional and metropolitan services. Being a state-wide initiative, a steering committee was established that consisted of nursing and midwifery leaders from regional and metropolitan areas. The steering committee was briefed on the reasons for the exchange, which included improved networking, communication and collaboration between health services; and fostering leadership and mentorship across diverse practice locations. As a new initiative to Queensland Health, it was important to evaluate the program with an intention of embedding it within the Queensland Health's early career support and rapid specialisation initiatives [17–20].

## Aims of the program

The aims of this study were to:

- Evaluate the efficacy and sustainability of NMEP;

- Develop a formal pathway for ongoing implementation across the health services;

- Explore the perceptions of key stakeholders of the exchange program;

- Identify if there are similar models that have been trialled in other countries and settings; and

- Gain a consensus from the managers of health services as to what they view as a sustainable model [21].

The research questions for analysis were:

1. Can exposure to clinical practice in alternate settings change future employment intentions as viewed by the nurses and midwives?

2. Is there evidence of:

    a. increased job satisfaction, and reduced burnout and job strain, amongst nurses and midwives who have completed an exchange placement?

    b. self-reported confidence in relation to clinical and professional practice?

    c. job and community embeddedness in practice?

3. Is NMEP financially sustainable in the long-term?

4. What is a sustainable model for NMEP, as viewed by experts?

## Method

A mixed methods approach to the project was adopted within a pragmatic framework to allow for the exploration of unknown variables [22, 23]. Pragmatic analysis allows for both quantitative and qualitative paradigms to be combined in a way that allows for the analysis of social phenomena, in real world situations that have not been fully explored [24]. By studying the narrative of nurses who have participated in the exchange, and combining this with statistical analysis, the barriers and enablers to NMEP are highlighted which will assist in reviewing and refining the program for future sustainability.

The project was undertaken in three stages.

*Stage One* was an online survey that was sent out to nurses and midwives who had participated in the exchange program. A comparison group was added; nurses who did not participate in the exchange program. The survey consisted of validated questionnaires to examine job strain, turnover intention, embeddedness, burnout and job satisfaction. Participants were asked to complete the surveys at the beginning of the exchange, during the exchange and then again on completion of the exchange. The participants were sent a link to the survey by the program coordinators and ask to complete at a time and place convenient to them. Participants from the exchange cohort were asked to create a unique identifier for use across the three surveys. Nurses who did not participate in exchange completed the survey once only. Table 1 provides a summary of the surveys used. Free text space was provided for participants to discuss their views of the program. These data were intended to be analysed statistically using generalised linear mixed models to examine time-related change and the narratives were examined thematically [25].

*Stage Two* was an integrative review to explore rural and remote nursing/midwifery recruitment and retention factors and to explore similar programmes or initiatives internationally (*publication under review*).

*Stage Three* was a Delphi inquiry that sought the views of the executive directors of nursing working across the state of Queensland. The Delphi [26] is a structured communication technique initially developed as a systematic, interactive forecasting method. In this study, the Delphi comprised a three-round combination of open and closed questions with mixed methods analysis, with the aim of achieving consensus on a sustainable NMEP model. Leaders in the field were from a consenting panel of executive directors of nursing from Queensland Health who were used to provide discussion and the exploration of ideas based on expert knowledge and experience.

**Table 1. Questionnaire used in the survey.**

| Construct Measured | Name of Measure | Description |
|---|---|---|
| Demographic questions | Related to age, location, experience | Establishment of context in normal practice |
| Questions related to NMEP experience | Questions aimed at finding out how well the exchange program worked for the participant (Likert scale and free text) | Questions aimed at collecting data related to the efficacy of the exchange program |
| Burnout | Burnout Measure–Short Version | 10-item version of the original 21-item scale. Example item: 'Difficulties sleeping'. *Malach-Pines, A. (2005). The Burnout Measure, Short Version. International Journal of Stress Management, 12(1), 78–88. doi*:10.1037/1072-5245.12.1.78 |
| Job Strain | General Health Questionnaire | A 12-item measure that captures general psychological distress using a 4-point Likert Scale. Example item: 'Felt constantly under strain'. *Golderberg, D., & Williams, P. (1988). A user's guide to the General Health questionnaire. Windsor, UK: NFER-Nelson.* |
| Job Satisfaction | Abridged Job in General | 8-item scale, a short version of the previous Job in General Scale. Russell, S. S., Spitzmüller, C., Lin, L. F., Stanton, J. M., Smith, P. C., & Ironson, G. H. (2004). Shorter can also be Better: The *Abridged Job in General Scale. Educational and Psychological Measurement, 64(5), 878–893. doi*:10.1177/0013164404264841 |
| Job and Community Embeddedness | Job Embeddedness Measure | Questions that examine the Fit, Links, and Sacrifice elements that contribute to the construct of embeddedness, reflected by both job-based and community-based factors. Example item: 'I feel like I am a good match for this organisation'. *Lee, T.W., Mitchell, T.R., Salbynski, C., Burton, J.P., & Holtom, B.C. (2004). The effects of job embeddedness on organisational citizenship, job performance, volition absences, and voluntary turnover. Academy of Management Journal, 47 (5), 711–722* |
| Attrition | Three-item Turnover Intention Scale | Three items using a five-point Likert scale measures how often respondents consider leaving their occupation, and likelihood of leaving their occupation in the future. Example item: 'How likely is it that you would leave your organisation in the next year? *Jaros, S. (1997). An assessment of Meyer and Allen's (1991) three-component model of organizational commitment and turnover intentions. Journal of Vocational Behavior. 51(3), 319–337.* |

## Ethics

Ethics approval was submitted through Darling Downs Hospital and Health Service Human Research Ethics Committee. Ethics exemption was granted as a Service Evaluation LRN/18/QTDD/44510 on the 11/09/18.

## Evaluation and results

The approach to the NMEP review and the results are discussed under each stage of the study.

### Stage one—The participant (nurses and midwives) survey

**Quantitative study.** Fifty-three nurses/midwives participated in the quantitative study (Female $n = 48$, Age $M = 36.69$ years ($SD = 12.12$)). Most participants worked in a metropolitan region as their regular location of work ($n = 28$), with rural ($n = 13$), and regional/remote ($n = 12$) participants being less common in the sample. Most of the participants role was a Registered Nurse ($n = 36$). A Bachelor's Degree was the most common highest qualification ($n = 31$), and most participants worked in a permanent full-time (38 hours per week) role ($n = 28$). Of the 53 participants 24 participants were in the exchange program, while 29 were comparison group participants. Nurses/midwives in the exchange program most commonly had a three-month exchange period ($n = 18$).

*Design and data analysis.* A cross-sectional between-groups design was employed for the quantitative analyses. As the quantity of participants in the exchange program who provided

multiple data points was very small ($n = 6$), the analysis plan was modified to focus on between-groups differences between the exchange participants and the comparison group due to statistical power being untenable for longitudinal analyses. For the exchange participants who presented multiple data entries, the last recorded response by the participant was used for the comparison to avoid non-independence of data and reflect a post-intervention response. Bayesian estimation of the mean score differences between groups (akin to an independent groups *t*-test using a null-hypothesis significance testing approach) was performed using the *BEST* package [27] in *R* software [28]. When conducting the analyses via BEST, chain lengths between $5e^4$ and $5e^5$ were specified per analysis until evidence of convergence for each model parameter via inspection of the $\hat{R}$ coefficient (i.e., $\hat{R} < 1.01$), was ascertained. Highest Density Interval (HDI) boundaries containing 95% of the parameter distribution (e.g., estimates of the mean difference in scores between the exchange group and the comparison group) were examined against a value of zero falling between the boundaries (i.e., no difference between groups). Descriptive statistics (correlations and coefficients of central tendency) were calculated using the *psych* package [29] and syntax published by [30].

*Statistical analysis.* Bayesian estimated correlations and coefficients of central tendency are presented in Table 2. Mean and standard deviations for each variable reflected a summed total across the variable's items. When examining $\omega_h$ reliability for each variable, one item was dropped from the summed total score of the embeddedness measures of organisational fit and organisational sacrifice due to poor loading on each measure's general factor. No further evidence of poorly-loading items was noted following these removals. Most measures appeared to be sufficiently reliable, although the occupational attrition intentions, sacrifice-based community embeddedness, and the wellbeing measures suggested potential reliability concerns (see Table 3). As the potential for attenuated correlations was not considered concerning due to the mean-difference-based analyses in the forthcoming section, the analyses continued in light of these findings.

**Table 2. Correlations, reliability, and central tendency coefficients for measured variables (n = 53).**

| | 1 | 2 | 3 | 4 | 5 | 6 | 7 | 8 | 9 | 10 | 11 | 12 | 13 |
|---|---|---|---|---|---|---|---|---|---|---|---|---|---|
| **1. JobSat** | 0.83/0.70 | | | | | | | | | | | | |
| **2. TI** | -0.39 | 0.87/0.86 | | | | | | | | | | | |
| **3. OA** | -0.29 | 0.57 | 0.65/0.60 | | | | | | | | | | |
| **4. Burnout** | -0.28 | 0.52 | 0.54 | 0.89/0.70 | | | | | | | | | |
| **5. FitCom** | 0.07 | -0.40 | -0.17 | -0.21 | 0.96/0.93 | | | | | | | | |
| **6. SacCom** | -0.07 | -0.18 | 0.15 | -0.11 | 0.74 | 0.64/0.58 | | | | | | | |
| **7. FitOrg** | 0.44 | -0.52 | -0.25 | -0.49 | 0.39 | 0.25 | 0.75/0.68 | | | | | | |
| **8. SacOrg** | 0.38 | -0.57 | -0.31 | -0.41 | 0.50 | 0.29 | 0.74 | 0.88/0.73 | | | | | |
| **9. GHQ** | -0.15 | 0.57 | 0.46 | 0.58 | -0.11 | -0.04 | -0.25 | -0.09 | 0.83/0.48 | | | | |
| **10. Leader** | -0.10 | 0.04 | -0.03 | -0.04 | 0.05 | 0.04 | -0.02 | -0.01 | -0.28 | NA | | | |
| **11. UndRural** | -0.03 | 0.35 | 0.22 | 0.03 | -0.23 | 0.16 | -0.16 | -0.24 | 0.04 | 0.15 | NA | | |
| **12. UndMetro** | -0.22 | 0.06 | -0.10 | 0.17 | 0.06 | -0.07 | -0.16 | -0.15 | -0.06 | 0.45 | -0.11 | NA | |
| **13. Network** | -0.05 | 0.34 | 0.20 | 0.18 | -0.04 | 0.18 | -0.13 | -0.13 | 0.07 | 0.33 | 0.35 | 0.34 | NA |
| **M** | 20.67 | 7.49 | 5.75 | 29.71 | 26.16 | 14.80 | 25.70 | 34.84 | 10.04 | 7.75 | 7.29 | 3.96 | 6.83 |
| **SD** | 4.64 | 3.53 | 2.33 | 9.04 | 7.53 | 3.32 | 4.46 | 9.02 | 4.33 | 1.33 | 2.10 | 1.07 | 2.02 |

$\alpha/\omega_h$ are presented along the diagonal for each variable. NA = Not available due to one or two items forming the measure. JobSat = Job satisfaction. TI = Turnover intention. OA = Occupational attrition intention. FitCom = Embeddedness Fit (Community). SacCom = Embeddedness Sacrifice (Community). FitOrg = Embeddedness Fit (Organisation). SacOrg = Embeddedness Sacrifice (Organisation). GHQ = Global Health Questionnaire. Leader = Self-rated leadership. UndRural = Understanding rural clients and practices. UndMetro = Understanding metropolitan clients and practices. Network = Perceived support.

**Table 3. Mean differences and highest density intervals for comparison and exchange group differences (n = 53).**

| Variable | Comparison ($\bar{x}_1(\sigma_1)$) | Exchange ($\bar{x}_2(\sigma_2)$) | $\bar{x}_1 - \bar{x}_2$ (95% HDI)[a] | d (95% HDI)[b] |
|---|---|---|---|---|
| Job Satisfaction | 22.43 (2.61) | 21.71 (3.03) | 0.73 (-1.36, 2.84) | 0.27 (-0.48, 1.04) |
| Turnover Intention | 6.53 (3.66) | 8.40 (3.36) | -1.87 (-3.90, 0.18) | -0.54 (-1.14, 0.04) |
| Occupation Attrition | 5.42 (2.33) | 6.07 (2.50) | -0.65 (-2.08, 0.73) | -0.28 (-0.84, 0.31) |
| Burnout | 29.62 (7.20) | 29.68 (11.36) | -0.06 (-5.64, 5.61) | -0.01 (-0.60, 0.56) |
| GHQ | 10.34 (3.73) | 9.44 (4.94) | 0.90 (-1.71, 3.51) | 0.21 (-0.39, 0.80) |
| Leadership | 7.74 (0.99) | 7.97 (0.81) | -0.23 (-0.92, 0.33) | -0.24 (-0.86, 0.37) |
| Understand Rural | 6.80 (2.30) | 8.02 (1.73) | -1.23 (-2.40, -0.05) | -0.61 (-1.20, -0.03) |
| Understand Metro | 3.98 (1.03) | 4.16 (0.74) | -0.18 (-0.75, 0.37) | -0.19 (-0.78, 0.44) |
| Network Adequacy | 6.88 (1.94) | 7.11 (1.62) | -0.22 (-1.26, 0.87) | -0.13 (-0.71, 0.48) |
| Embeddedness | | | | |
| Comm. Fit | 25.23 (7.23) | 28.55 (6.64) | -3.32 (-7.48, 0.94) | -0.49 (-1.10, 0.16) |
| Comm. Sacrifice | 13.79 (3.24) | 16.30 (2.85) | -2.50 (-4.33, -0.71) | -0.83 (-1.47, -0.23) |
| Org. Fit | 25.55 (4.94) | 26.17 (3.97) | -0.62 (-3.19, 2.03) | -0.14 (-0.72, 0.45) |
| Org. Sacrifice | 36.32 (8.74) | 34.31 (8.49) | 2.01 (-3.21, 7.38) | 0.23 (-0.38, 0.83) |

[a] Mean difference between comparison ($\bar{x}_1$) and exchange ($\bar{x}_2$) scores on each variable, with 95% Highest Density Interval of parameter estimates. A positive score means the comparison group had a higher mean score, while a negative score means the exchange group had a higher mean score.

[b] d effect size estimate, with 95% Highest Density Interval of parameter estimates.

Comm. = Community. Org. = Organisational.

*Mean difference analyses.* As outlined in Table 3, differences between the comparison and exchange groups varied on two of the work-related measures. The reported degree of understanding of rural client and health issues differed between groups, with nurses/midwives in the exchange group reporting a higher score on this measure, which was accompanied with a moderate-to-large effect size estimate (d = 0.61). The largest difference between the comparison and exchange group was on the embeddedness measure that reflected sacrifice perceptions within the nurse's community if they had to leave it to work elsewhere. Nurses/midwives in the exchange group reported higher scores on their perceptions of aspects of their home community that would be lost if they had to leave, which was accompanied with a large effect size (d = 0.83). The remaining between-group comparisons, however, had 95% HDI boundaries that encompassed zero, which therefore suggested no interpretable difference between the exchange and comparison group. Consequently, limited evidence of comparison and exchange group differences on the attitudinal measures was present in these findings.

**Thematic analysis.** The survey sent out to the nurses and midwives who participated in NMEP had space for free text comments. This data was subject to thematic analysis, in which three of the research members collated the feedback into major themes. When considering their job in general, 96% said it was good, with 84% saying their job made them content and 89% said their job was enjoyable, all with a good understanding of their workplace and clinical practice. Fig 1 below shows participants likelihood of leaving their job in nursing. Some comments related specifically to the exchange program while others related to nursing in general.

> *I like nursing and I can't think of a better job. I feel like I am not growing in my current role that's why I did the exchange program.*

> *Due to a current situation within the workplace, sometimes I find myself considering other places of work at the end of my contract due to the effect it has on staff. However, the potential*

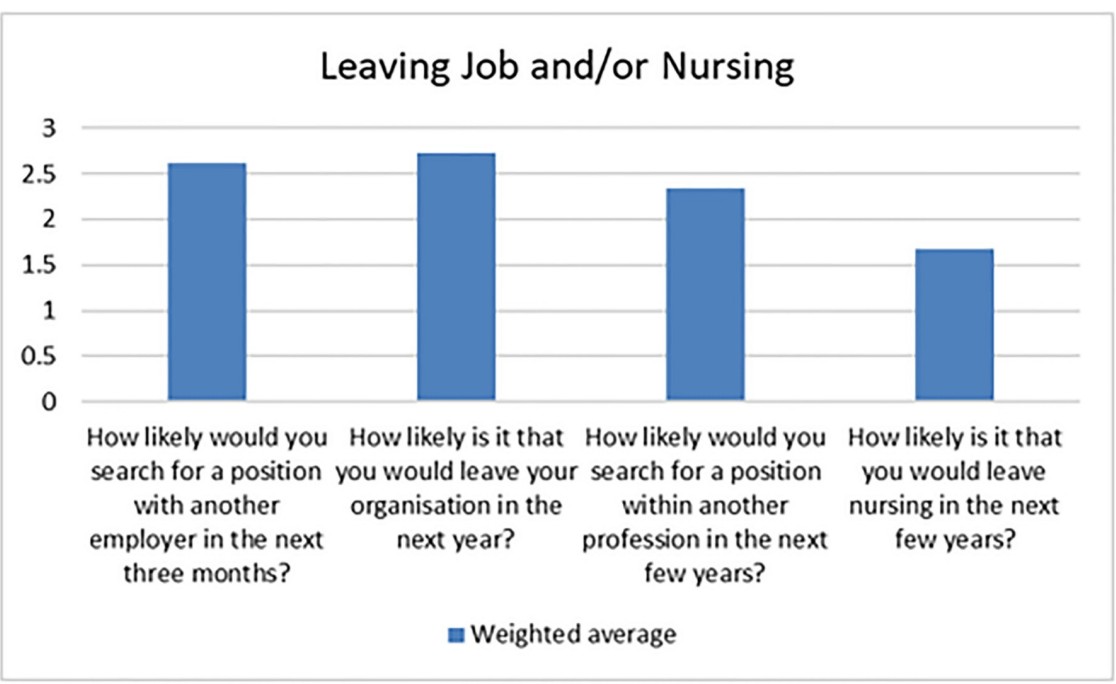

**Fig 1. Likelihood of leaving job or nursing.**

*that this place offers is enormous, once things get sorted out of course. To be in this role at such a junior level is amazing, and I do love my job here.*

Although the option of free text was not overly utilised, participants responses indicated some concern over the workplace environment. One mentioned bullying by senior nurses, whilst other participants indicated unfair work practices such as:

*Some team leaders and staff members are rude or show favouritism to their friends at work which [is] unfair and upsetting.*

Sixteen nurses/midwives indicated thoughts about leaving their current employment with ten nurses/midwives suggesting they were undecided. One nurse said that she had "*already obtained employment in organisation of which I am exchanging to*". One nurse said they intended leaving nursing altogether, whilst others were undecided (n = 11). Comments suggesting that the workplace was not so satisfying include,

*Currently I love my job, I just don't love my workplace at times and the support I receive from other staff.*

*Lots of potential here, just feel undervalued at times for the amount of effort I expend, however.*

*I really hope that with the change of environment, new workmates, new challenges I will be able ignite my enthusiasm, and get my confidence back.*

Despite some participants suggesting they were going to leave either their job or nursing, most nurses/midwives did not indicate feeling trapped, helpless or worthless, although 88% indicated feeling tired (sometimes, often, very often or always).

The majority of nurses/midwives indicated that mentorship in clinical practice was important, with one indicating that mentoring in the exchange program requires improvement.

*Mentoring in my normal job in [Metro health service] has been beneficial to my career and practice. I have not received much mentoring since starting the exchange.*

When asked about their leadership and experience, most felt confident about their leadership skills. One nurse said,

*There is definitely room for improvement and working in a new section/higher roles in your workplace makes you feel back to basics in some ways.*

### Stage two—The literature review

An integrative review was undertaken to identify key factors that influence recruitment and retention of the rural and remote nursing and midwifery workforce. Detailed information has been submitted to a peer reviewed journal.

### Stage three—The Delphi

The first round Delphi questionnaires comprised a combination of open and closed questions using Survey Monkey™. Closed questions were used, asking panellists to specifically rate through a 5-point Likert system some component of the NMEP sustainability model; panellists were asked to explain their opinions. The second-round questionnaire asked further questions on new issues that emerged from responses to the previous open questions, plus iterated closed questions. Feedback on the opinions of panellists on the first two rounds along with summaries of the written arguments given by panellists pre-empted Round 3, a survey where synthesized themes were incorporated into a Likert-type scale, and the leadership panel participants were asked to rate and validate responses in order to achieve consensus. Participants were asked to rate statements, which were both positively and negatively formulated, using a five-point Likert scale, effectively re-ranking components from strongly disagree (1) to strongly agree (5), with the option to include comments if desired. The weighted mean and standard deviation of all answers were computed for each item (separately for each round) as a measure of the spread in responses across participants and was used to calculate the change in each item's variability between rounds. The overall agreement among the leaders was determined with the intraclass correlation coefficient (ICC), with consensus and stability tested by 2-way random ANOVA with absolute agreement. The ICC is interpreted as follows: ≤0.40, poor consistency or large variation in opinion; 0.41–0.74, acceptable consistency; and ≥0.75, good consistency.

The research question was, *what is a sustainable model for NMEP, as viewed by experts*?

**Participants.** Fifteen executive directors of nursing from 16 Queensland health services participating in the exchange program were invited to participate.

*Round one*. The survey contained nine open ended questions and 17 items for rating. The focus of round one was to *gain information about workforce recruitment and retention related to the NMEP* toward developing a sustainability model. The components of the NMEP were transferred to items and rated regarding importance.

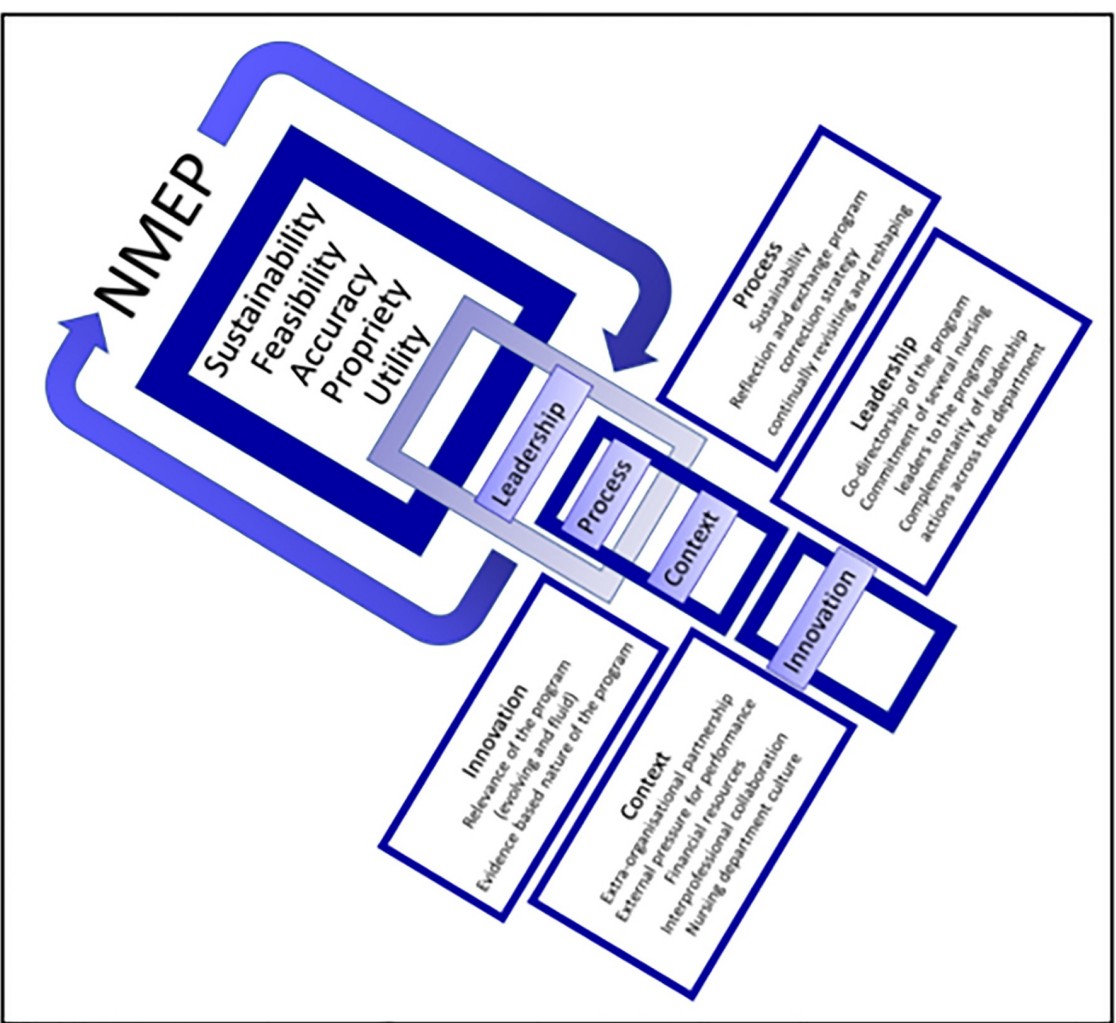

**Fig 2. NMEP sustainability model.**

*Round two*. Round Two included a sustainability model adapted from a conceptual framework [31, 32]. The model contains four categories of factors proposed to influence sustainability, these are: innovation, context, leadership, and process. Innovation factors pertain to the nature of the program that is new to the organisation or to a group of individuals at the time of adoption. Contextual influences are due to the environment, setting, situation, or conditions within which the innovation is implemented. Leadership is the formal or informal manager(s) or organiser(s) of a group, with certain authorities, attributes, and actions that influence other people. Process factors refer to series of events, strategies, or activities that lead to a particular result. Refer to Fig 2, *The NMEP conceptual model for sustainability*, which can be found in the Discussion. Round Two survey contained five open ended questions. The focus of round two was to *gain information about how the conceptual model for sustainability relates to the NMEP* toward developing a sustainability model.

*Round three*. In this round the themes and major points from Round 2 were converted to quantitative items. Round three contained 28 items for rating. The focus of round three was to gain information *about the importance of different concepts related to the NMEP* toward developing a sustainability model.

**Table 4. Rate how you regard the following components of the NMEP program toward the successful recruitment of nurse and midwives.**

| Intraclass correlation .776 [good consistency and agreement] | Mean | Std. Deviation |
|---|---|---|
| ANOVA F = 9.573 (9,63 df), p ≤0.001 | | |
| Marketing of the NMEP improves visibility and opportunity | 4.3 | .67 |
| Mentoring using the mentoring toolkit incentive | 4.1 | .56 |
| New experience and a change incentive | 4.1 | .87 |
| Financial incentives | 4.1 | .73 |
| Education and learning incentives | 4.0 | .66 |
| Moving from a metropolitan location to a rural or regional location. | 3.9 | .56 |
| Time the exchange is organised for | 3.8 | .91 |
| Moving from a rural or regional location to a metropolitan location. | 3.6 | 1.26 |

*Results.* Ten participants completed the survey. There was strong consensus in ratings and high agreement and consistency between panel members with retention (Table 5) showing higher consistency and agreement. Marketing of the NMEP was the highest scoring item for both recruitment and retention. The lowest scoring item in regard to both recruitment and retention was "Moving from a rural or regional location to a metropolitan location." Refer to Tables 4 and 5 for the rating questions with the highest weighted mean scores descending. The open-ended questions used content analysis for major themes. In regard to *recruitment*, the most frequent keyword were "opportunities" and "support". Refer to Table 6 for the narrative answers to the open-ended questions re *recruitment*. In regard to *retention* the most frequent keyword were "opportunities" and "work culture". Refer to Table 7 for the narrative answers to the open-ended questions re *retention*.

**Round two.** Ten participants completed the survey, these were the participants who had direct involvement with the program. The narratives derived from the open-ended questions aligned to each concept (Table 8).

**Round 3.** Five out of the original 15 participants completed the survey. There was strong consensus in ratings and high agreement and consistency between panel members with CON-TEXT RELATED FACTORS (Table 11) showing highest consistency and agreement. The highest mean weighted scores (most important) items were under LEADERSHIP, "*The program is reliant on senior leaders being part of the program to ensure that it is accepted at their facilities and that staff that want to engage in the program are empowered to do so.* And INNO-VATION, "*The program should be reviewed in line with evolving research and industry changes*

**Table 5. Rate how you regard the following components of the NMEP program toward the successful retention of nurse and midwives.**

| Intraclass correlation .895 [good consistency and agreement] | Mean | Std. Deviation |
|---|---|---|
| ANOVA F = 4.479 (9,63 df), p ≤0.001 | | |
| Marketing of the NMEP improves visibility and opportunity | 4.2 | .63 |
| Financial incentives | 4.2 | .63 |
| Mentoring using the mentoring toolkit incentive | 4.1 | .56 |
| Education and learning incentives | 4.1 | .56 |
| Moving from a metropolitan location to a rural or regional location. | 4.0 | .81 |
| New experience and a change incentive | 4.0 | .81 |
| Time the exchange is organised for | 3.9 | .87 |
| Moving from a rural or regional location to a metropolitan location. | 3.7 | .82 |

**Table 6. Strategies for the successful recruitment of nurses and midwives' narratives and word cloud.**

*"dedicated and targeted recruitment team to assist NUMs to process applications in a timely manner"*

*"HHS covering regional, rural and remote: Appropriate postgraduate support and skills development A positive workforce culture. Development and career progression opportunities Supported on boarding of new staff Support with accommodation/relocation."*

*"In rural remote important for staff to have an opt out ability so that they don't have to give up a job at the coast or metro until they determine that they like it in rural and remote. Education pathways and opportunities that tie staff to an area. Social integration and onboarding to connect them to a social network. Good quality accommodation that is provided free. Rural and remote incentives as a value add to normal conditions to attract."*

*"Longer term view in terms of skills needed for the future esp in mental health and community. need to look at alternative roles and methods of training delivery"*

*"Need to be able to interchange staff and give training opportunities"*

*"Positive workforce, opportunity, culture"*

*"Robust graduate program, invest in succession planning"*

*"Rural and regional—availability to experienced staff at short notice to provide backfill for emergent leave, support of early career nurses in specialist areas."*

*"There needs to be a financially supported pathway to support nurses transitioning into remote practise*

*to ensure relevance"* and CONTEXT RELATED FACTORS, *"Nursing cultural issues should be enhanced so that an exchange is a growth experience that builds staff abilities through diversity and building of networks across the state as well as sharing of knowledge."*

Refer to Tables 9–13 for the items with the highest weighted mean scores descending within the concepts that form the validated *Conceptual model for sustainability for the NMEP program*. All items contributed to the model, except for those that rated below a Mean < 4.00.

## Summary of findings

This study set out to review NMEP, an innovative human resource initiative of metro/rural nursing and midwifery exchange, aimed at improving the visibility of nursing in regional, rural and remote areas, providing professional development opportunities for nurses and midwives, and promoting a better understanding of metro and rural nursing and midwifery to support collaborative practice. The evaluation sought to identify whether this pilot program could be sustainable, and if there was evidence of increased job satisfaction. In the systematic

**Table 7. Strategies for the successful retention of nurses and midwives' narratives and word cloud.**

*"A lot retention in remote is about the living, currently not the best access to great accommodation, often sharing."*

*"career pathways social integration Quality free accommodation Financial incentives to stay Interesting work"*

*"HHS covering regional, rural and remote: Positive workforce culture development and career progression opportunities Flexible working/family friendly Opportunities to work throughout whole HHS particularly for rural/remote employees"*

*"more flexible working arrangements, step on step off models of training, opportunity to shadow more senior roles"*

*"Positive culture, workforce and opportunity."*

*"Regional and rural—Professional development opportunities, rotational opportunities to maintain clinical skills"*

*" Availability of permanent positions if candidates decide they want to stay. Staff that have undertaken the exchange program then having an avenue to be on managed secondment lists for future opportunities and a unit to coordinate this."*

*"The culture of the unit and the organisation and work environment have an impact on retention. Educational opportunities are also important to nurses."*

*"Use a capability framework, mentoring and appropriate education"*

**Table 8. Narrative from open-ended questions.**

| Theme | Comments from Participants |
|---|---|
| Leadership<br>Commitment of several nursing leaders to the program over a consistent period of time, [2] co-directorship of the program working closely together between workforce needs, practice and research, [3] complementarity of leadership actions across the various levels of Health. For the NMEP to be sustainable, this needs to be evident. | • Leadership is key to driving the success of this project<br>• Program requires strong leadership from dedicated leaders to ensure viability and continual growth.<br>• The program is reliant on senior leaders being part of the program to ensure that it is accepted at their facilities and that staff that want to engage in the program are empowered to do so.<br>• Very important to the success<br>• Yes Leadership is key to embedding NMEP into the culture of HSS's.<br>• All nursing leaders should be committed to the program |
| Innovation<br>Innovation is a core concept of sustainability. Innovation in this context is the [1] Relevance of the program, that addresses the issues in a bigger picture manner and are continually evolving. [2] Nature of the program, there has to be a reason why it evolves in a particular way, relying on broad evidence, rather than just one source (i.e. finances only) For the NMEP to be sustainable, this needs to be evident. | • Innovation can come from the learning's within these environments and can be transferred Innovative models of care and research opportunities can ensure the programs sustainability<br>• Staff respond well to out of the box ideas<br>• The aim of the program is to provide opportunities for nurses to increase their skills and knowledge in different health settings.<br>• The program needs to be responsive and flexible, it is an iterative process that should evolve to reach its potential<br>• The program should be reviewed in line with evolving research and industry changes to ensure relevance. Whilst financials are important all factors that enable success and measures of success need to be equally evaluated and considered.<br>• The program has the ability to evolve and offer new products or offerings utilising the existing staff and the exchange alumni that have undertaken an exchange.<br>• Program could be expanded within current resources to offer a service that can match casuals and interested staff that want secondments to temporary vacancy |
| Context Related Factors<br>Related factors are core concepts of sustainability. Context factors in this construct are the [1] Extra-organisational partnerships—interrelated professional partnerships that achieve more together [2] External pressure for performance—i.e. from consumers and government bodies [3] Financial resources—limited budget to achieve desired outcomes [4] Nursing Culture—push and pull of nursing culture in a political and macro health perspective. For the NMEP to be sustainable, there needs to be evidence that these factors are being managed for a positive outcome. | • HHS s are generally enthusiastic to support programmes, but budget constraints are often prohibiting<br>• In health these factors are always evident to do more with less and value for money for the service we deliver.<br>• I think there needs to be consideration of the financial impacts.<br>• The program will not be sustained unless the management staff of the program are maintained as a funded unit.<br>• As soon as it becomes a user pays system then the costs will outstrip benefits. Staff being released to do the program is at times also a problem and more support is required to ensure that staff are empowered to apply and go.<br>• Nursing cultural issues should be enhanced so that an exchange is a growth experience that builds staff abilities through diversity and building of networks across the state as well as sharing of knowledge.<br>• Utilising existing resources effectively is key |

(*Continued*)

**Table 8.** (Continued)

| Theme | Comments from Participants |
|---|---|
| Process Related Factors<br>Process related factors are a core concept of sustainability. Process related factors in this context are the [1] Reflection and program-correction. This refers to iterations over time of leaders' deliberate efforts to learn from program experiences and, in response, to try to implement continued improvements to the program. For the NMEP to be sustainable, this needs to be evident. | • excellent program staff really loved it<br>• Process related factors for improvement have most likeable been limited to the processes of the exchange and to marketing the program.<br>• There has been limited use of collected information along the program that have leveraged the use of experience measures.<br>• The lessons learned need to be shared<br>• The program has been well received but frequent evaluation is required to maintain standards.<br>• Buy in and evaluation |

review undertaken for this evaluation, there were no other programs that reflected the intention of this work.

Overall, the findings from NMEP were positive, with both nurses/midwives and leaders viewing the project as important for professional development, innovation and retention of staff. The Delphi provided rich and valuable information around NMEP's sustainability and value for the nursing workforce. Leaders identified key factors that support the nursing workforce through improving visibility of the nursing workforce in rural and regional areas, providing opportunities for advancing knowledge and practice, and establishing long term collaborative opportunities between metro and rural nursing, with some excellent responses to how this should be developed and carried forward. Key points identified are captured in Fig 2.

Financial resources for the project were seen as negatively affecting the future of NMEP, whilst leadership was viewed as essential, requiring a commitment of nursing leaders to create a strategic alliance between metro and rural health services. The program was also seen as responsive to a dynamic healthcare need, a matter which has been raised as an essential element to a future, responsive health care workforce [33, 34]. NMEP as a pilot was acknowledged with the process to sustainability being iterative as the program is further developed.

**Table 9. Rate the following components regarding the success and sustainability of the NMEP program.**

| Intraclass correlation .763 [good consistency and agreement] ANOVA F = 3.818 (3,18 df), p ≤0.05 | Mean | Std. Deviation |
|---|---|---|
| Effective marketing of the exchange program needs to be formally organised | 4.40 | .89 |
| There needs to be a financially supported pathway to support nurses transitioning into remote practice as part of the exchange program | 4.40 | .54 |
| More than one opportunity to exchange, placed on waiting lists for secondment. | 4.25 | .50 |
| There needs to be a dedicated funded unit that can administer an effective exchange program. | 4.20 | .83 |
| Its important for staff exchange to rural and remote to have an opt out ability so that they don't have to give up a position. | 4.20 | .44 |
| Availability of permanent positions if candidates decide they want to stay in rural and remote. | 4.00 | 1.22 |
| Senior and experienced staff should be encouraged with incentives to participate in the exchange program | 3.60 | 1.14 |

**Table 10. Rate how you regard the following components in regard to LEADERSHIP influencing the success and sustainability of the NMEP program.**

| Intraclass correlation .763 [good consistency and agreement] ANOVA F = 5.841 (4,16 df), p ≤0.05 | Mean | Std. Deviation |
|---|---|---|
| The program is reliant on senior leaders being part of the program to ensure that it is accepted at their facilities and that staff that want to engage in the program are empowered to do so. | 4.60 | .54 |
| Leadership is key to embedding NMEP into the culture of Hospital and Health Services | 4.40 | .54 |
| Program requires strong leadership from dedicated leaders to ensure viability and continual growth. | 4.20 | .83 |
| Availability of permanent positions if candidates decide they want to stay in rural and remote. | 4.00 | .70 |
| All nursing leaders should be committed to the program | 3.60 | 1.14 |

**Table 11. Rate how you regard the following components in regard to INNOVATION influencing the success and sustainability of the NMEP program.**

| Intraclass correlation .777 [good consistency and agreement] ANOVA F = 4.679 (4,16 df), p ≤0.01 | Mean | Std. Deviation |
|---|---|---|
| The program should be reviewed in line with evolving research and industry changes to ensure relevance. | 4.60 | .54 |
| The program needs to be responsive and flexible, it is an iterative process that should evolve to reach its potential. | 4.40 | .54 |
| The NEMP program has the ability to evolve and offer new products or offerings using existing staff and the exchange alumni that have undertaken an exchange. | 4.20 | .83 |
| Innovative models of care and research opportunities can ensure the programs sustainability. | 4.00 | .70 |
| Innovation can come from the learning in new environments and can be transferred to other nurses/midwives or health services. | 4.00 | .70 |
| Comment: The more flexibility with the program the better as it will then be an attractive option | | |

Leaders identified that permanent positions should be available so that if nurse/midwives wanted to transfer to a regional or rural area, this can be expedited.

There are three main themes emerging from the evaluation: Embeddedness, burnout and financial support. Each of these themes is described in more detail under the relevant headings below.

**Table 12. Rate how you regard the following components in regard to CONTEXT RELATED FACTORS influencing the success and sustainability of the NMEP program.**

| Intraclass correlation .872 [good consistency and agreement] ANOVA F = 7.228 (4,16 df), p ≤0.01 | Mean | Std. Deviation |
|---|---|---|
| Nursing cultural issues should be enhanced so that an exchange is a growth experience that builds staff abilities through diversity and building of networks across the state as well as sharing of knowledge. | 4.60 | .54 |
| Using existing resources effectively is key to sustainability. | 4.40 | .89 |
| HHS budget constraints need to be supported and managed so that those who are enthusiastic can have opportunities to exchange. | 4.20 | .83 |
| Staff being released to do the NMEP can be problematic; more support is required to ensure that staff are empowered to apply and go. | 4.20 | .83 |
| The NMEP program will not be sustained unless the management staff of the program are maintained as a funded unit. | 4.20 | .83 |

**Table 13. Rate how you regard the following components in regard to PROCESS RELATED FACTORS influencing the success and sustainability of the NMEP program.**

| Intraclass correlation .869 [good consistency and agreement] ANOVA F = 7.143 (3,15 df), p ≤0.01 | Mean | Std. Deviation |
|---|---|---|
| Buy in and evaluation is key to sustainability | 4.50 | .57 |
| Using existing resources effectively is key to sustainability. | 4.40 | .89 |
| Evaluation and experience data both quantitative and qualitative needs to be collated to leverage improvements and sustainability. | 4.40 | .54 |
| Frequent evaluation of the NMEP is required to maintain quality standards. | 4.20 | .83 |
| The lessons learned from exchange participants and HHS need to be shared | 4.00 | .70 |
| Process related factors for improvement are likely to be limited if there are obstructions to the processes and marketing of the program. | 3.60 | 1.14 |

## Embeddedness

Job embeddedness can negatively affect staff retention and turnover [35–37]. In rural and remote nursing, connections with the community are a key factor to keeping staff [38]. The connection with their home community was evident, with nurses/midwives participating in the exchange expressing concerns over leaving their community and losing their connection with it. This finding is supported by recent work undertaken around rapid specialisation of the nursing workforce where a sense of belonging in the community was important [5, 9].

## Burnout

Job dissatisfaction is closely linked to work intensification, which in turn is linked to emotional exhaustion, cynicism and a lack of self-efficacy, all leading ultimately to burnout and cognitive dissonance [39–41]. In nursing, this has been shown to negatively affect patient outcomes [42, 43]. In rural and remote nursing locations, the added responsibility of making decisions onsite with the support of a geographically dislocated medical officer, can increase the sense of anxiety for a nurse, particularly when they are inexperienced in dealing with the wide range of clinical presentations [44]. Although burnout was not evident in the analysis, most nurses/midwives identified being tired. Nurses/midwives indicated their love of the job, but that it was the workplace environment that led to work dissatisfaction, a finding that has been repeatedly identified in international research related to work intensification [45–48]. The work by Hegney and colleagues [48] particularly relates to Queensland nurses drawn from a three yearly nursing workforce review in that state, in which nurses consistently reported a lack of support by managers and insufficient staff for the work at hand, resulting in exhaustion and not being able to complete care. The issue therefore, of nurses verbalising fatigue, should be of concern for managers, who should perhaps review their workforce and the workload allocations.

## Financial support

The leaders referred to the need for financial support for NMEP. Their view was that without a budget, maintaining the project would be problematic, for example,

> "*HHSs are generally enthusiastic to support programmes, but budget constraints are often prohibiting. In health these factors are always evident—to do more with less and value for money for the service we deliver*"

*The program will not be sustained unless the management staff of the program are maintained as a funded unit.*

This issue is not uncommon in health services, where projects that are funded for evaluation fail to attract ongoing funding. Reasons for this largely revolve around annualised budgets where projects that require more time to become embedded and cost effective, are given away for more immediate returns on investment, despite long term value being identified [49–51]. However, models of care have to change in response the push/pull factors of patients being moved much earlier from acute services into community and home based care as a result of increasing costs of chronic care [12, 33]. Projects such as NMEP should be carefully considered as part of the workforce change that needs to occur, in order to meet healthcare needs. Importantly, NMEP has the potential to support collaborative practice between metro and rural areas; given the need to enhance the skills and knowledge of the rural nursing and midwifery workforce, and the clinical reciprocity required for the success of integrated models of care [13, 52, 53].

## Discussion

No one program or strategy is going to be the panacea to the issue of rural and remote nursing and midwifery recruitment. The solution is more likely to lie in a multi-pronged approach that incorporates the nurse/midwife as a professional with the nurse/midwife as a community and family member, as well as providing opportunities for ongoing professional development.

The role of government must also be considered, and changes to current policies and processes may be necessary. Research conducted by Smith et al [54] showed that advanced practice nurses reported the regulatory system to be prohibitive and obstructive at times. The social factors that influence access to services remain key, but geographic barriers must be considered in context. Without considering the geographic barriers, additional interventions to promote access to services will inevitably be hindered. The complex interplay of competing factors in rural and remote nursing and midwifery are also highlighted by McCullough, Whitehead, Bayes, et al. [55] whose substantive theory of making compromises to provide primary health care in a remote setting clearly identifies the social, cultural and professional elements of rural and remote nursing and midwifery practice.

Strategies to encourage a suitably skilled workforce to rural and remote areas must also be developed with an interdisciplinary approach. Avenues for rural practice exist for medical staff [54] and for undergraduate nurses [56] and midwives, but little is available for nurses and midwives who are currently practising, with even more limited support for advanced practice nurses [57]. Exposure to rural and remote area nursing and midwifery practice should, ideally, be integrated into undergraduate studies, as is reflected in the existing literature [56] and there is still much work to be done in this space. However, undergraduate experiences should be the beginning of the overarching strategies and continue along the career pathway of nurses and midwives, including transition to practice [58].

Career progression and access to professional development opportunities can be limited for those nurses and midwives who live and work in rural and remote areas. Building understanding of the inherent challenges of maintaining the rural and remote workforce is not the sole responsibility of senior health department executives. Formal education and awareness programs should also be developed and implemented for low-middle level managers who may be best positioned to advocate for sustainable change. Professional development opportunities require leadership and commitment from managers to provide formal mechanisms of support to staff, yet the size and nature of rural and remote facilities can be a barrier to this [57,58].

Further, closer examination of extending the scope of practice of rural and remote practising nurses and midwives that recognises professional boundaries without being hobbled by them is required. Additionally, the community voice must remain central to the discussion for two main reasons. One, community members are the recipients of care and those whose quality of life is most affected and two, community acceptance and support is vital to retaining the rural and remote workforce for all disciplines. It is possible to maintain professional boundaries and the search for the best way to meet the community needs as identified by them, must continue with the priority being community health outcomes.

## Conclusion

Like all locally developed projects, NMEP was created in response to a problem, where staff saw the need to a) raise the visibility of nursing in rural and remote locations, b) encourage an exchange of staff and clinical expertise that would support better care and ultimately better patient outcomes, and c) provide an opportunity for metropolitan and rural nurses and midwives to collaborate more effectively, a matter important in the changing dynamic of care. The outcomes from NMEP show the potential for impact on workforce challenges from such programs and provide the evidence upon which to base considerations for future planning and sustainability. However, in order for this to occur, the ongoing support of leaders and committed funding over the long term is required.

## Conditions relative to the evaluation

This was a pilot study, undertaken to test the feasibility of an exchange program. Numbers within the exchange were small, based upon the capacity of the smaller rural service being able to physically and operationally accommodate the exchange. Consequently, the quantitative comparisons between exchange participants and comparison participants was potentially limited in the accuracy of the estimated highest density intervals, and therefore potentially the identification of measures that varied between the groups. As a counterpoint to this potential limitation, robust Markov Chain Monte Carlo methods [59] employed as part of the *BEST* package demonstrated model convergence evidence which strengthened our decision to interpret the highest density intervals as suggestive of no clear evidence of differences between the groups in most instances. A larger future sample, however, would assist in the accuracy of our coefficient estimates, therefore re-examination with a larger future sample is a recommended course of future research if the exchange program continues. Although nurses and midwives did not make use of the free text section of the survey, their responses support those of larger studies undertaken nationally and within the state of Queensland.

## Supporting information

**S1 File. COREQ checklist.**
(PDF)

**S2 File. CREDES checklist.**
(PDF)

## Author Contributions

**Conceptualization:** Amy-Louise Byrne, Clare Harvey, Diane Chamberlain, Adele Baldwin.

**Data curation:** Diane Chamberlain.

**Formal analysis:** Amy-Louise Byrne, Clare Harvey, Diane Chamberlain, Adele Baldwin, Brody Heritage, Elspeth Wood.

**Funding acquisition:** Clare Harvey.

**Methodology:** Clare Harvey, Adele Baldwin.

**Project administration:** Clare Harvey.

**Validation:** Amy-Louise Byrne.

**Writing – original draft:** Amy-Louise Byrne, Clare Harvey, Adele Baldwin.

**Writing – review & editing:** Amy-Louise Byrne, Clare Harvey, Diane Chamberlain, Adele Baldwin, Brody Heritage, Elspeth Wood.

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
