## [Decision Letter · Decision Letter 0]

11 May 2020

PONE-D-20-08725

Evaluation of a nursing and midwifery exchange between rural and metropolitan hospitals: a mixed methods study.

PLOS ONE

Dear Amy-Louise Bryne,

Thank you for submitting your manuscript to PLOS ONE. After careful consideration, we feel that it has merit but does not fully meet PLOS ONE’s publication criteria as it currently stands. Therefore, we invite you to submit a revised version of the manuscript that addresses the points raised during the review process.

We would appreciate receiving your revised manuscript by 15 June. . To enhance the reproducibility of your results, we recommend that if applicable you deposit your laboratory protocols in protocols.io, where a protocol can be assigned its own identifier (DOI) such that it can be cited independently in the future. For instructions see: http://journals.plos.org/plosone/s/submission-guidelines#loc-laboratory-protocols

We look forward to receiving your revised manuscript.

Kind regards,

Sharon Mary Brownie

Academic Editor

PLOS ONE

Additional Editor Comments:

Please carefully consider the recommendations tabled by the reviewers. Please prepare a table to describe how you have responded to each comment and please submit this with your revised manuscript. Questions have been raised about the statistical analysis within your manuscript. It is recommended that your obtain an expert statistical review prior to re-submission.

2. Please amend the manuscript submission data (via Edit Submission) to include author Elspeth Wood.

3. Please ensure that you refer to Figure 1 in your text as, if accepted, production will need this reference to link the reader to the figure.

Reviewers' comments:

Reviewer's Responses to Questions

**Comments to the Author**

1. Is the manuscript technically sound, and do the data support the conclusions?

Reviewer #1: Yes

Reviewer #2: Yes

2. Has the statistical analysis been performed appropriately and rigorously? 

Reviewer #1: Yes

Reviewer #2: I Don't Know

3. Have the authors made all data underlying the findings in their manuscript fully available?

Reviewer #1: Yes

Reviewer #2: Yes

4. Is the manuscript presented in an intelligible fashion and written in standard English?

Reviewer #1: Yes

Reviewer #2: Yes

5. Review Comments to the Author

Reviewer #1: 1- Abstract should be modified: increase the presentation of numerical results and cut-short the discussion or mention the conclusion only.

2- Methodology need to be focused and reproducible

3- Results are good

Reviewer #2: ABSTRACT: Under the introduction,the state were the study was carried out should be stated. The authors should clarify whether the last portion of the abstract is the conclusion or discussion.

introduction: This should be fused with the background as a large portion of the contents of the background(page 5 lines 98-121) is repeated in the introduction. PAGE 3,LINE 67 QECD SHOULD 1ST BE WRITTEN IN FULL.

6. PLOS authors have the option to publish the peer review history of their article (what does this mean?). If published, this will include your full peer review and any attached files.

Reviewer #1: No

Reviewer #2: No

---

## [Author Response · Author response to Decision Letter 0]

19 May 2020

Revised manuscript as per reviewers comments- see rebuttal letter

---

## [Editor Report · Decision Letter 1]

21 May 2020

Evaluation of a nursing and midwifery exchange between rural and metropolitan hospitals: a mixed methods study.

PONE-D-20-08725R1

Dear Dr. Amy-Louise Byrne,

We are pleased to inform you that your manuscript has been judged scientifically suitable for publication and will be formally accepted for publication once it complies with all outstanding technical requirements.

With kind regards,

Sharon Mary Brownie

Academic Editor

PLOS ONE

Additional Editor Comments (optional):

Authors have addressed recommendations made by the reviewers

---

## [Editor Report · Acceptance letter]

12 Jun 2020

PONE-D-20-08725R1 

Evaluation of a nursing and midwifery exchange between rural and metropolitan hospitals: a mixed methods study. 

Dear Dr. Byrne:

I'm pleased to inform you that your manuscript has been deemed suitable for publication in PLOS ONE. Congratulations! Your manuscript is now with our production department. 

Kind regards, 

on behalf of

Professor Sharon Mary Brownie 

Academic Editor

PLOS ONE